# Effects of Biotin on survival, ensheathment, and ATP production by oligodendrocyte lineage cells in vitro

Qiao-Ling Cui[1], Yun Hsuan Lin[1], Yu Kang T. Xu[1], Milton G. F. Fernandes[1], Vijayaraghava T. S. Rao[2]¤, Timothy E. Kennedy[1], Jack Antel[1]*

1 Montreal Neurological Institute, McGill University, Montreal, Quebec, Canada, 2 Medday Pharmaceuticals Inc., Boston, Massachusetts, United States of America

¤ Current address: McGill University, Montreal, Quebec, Canada
* jack.antel@mcgill.ca

**Data Availability Statement:** All relevant data are within the manuscript and its Supporting Information files.

## Abstract

Mechanisms implicated in disease progression in multiple sclerosis include continued oligodendrocyte (OL)/myelin injury and failure of myelin repair. Underlying causes include metabolic stress with resultant energy deficiency. Biotin is a cofactor for carboxylases involved in ATP production that impact myelin production by promoting fatty acid synthesis. Here, we investigate the effects of high dose Biotin (MD1003) on the functional properties of postnatal rat derived oligodendrocyte progenitor cells (OPCs). A2B5 positive OPCs were assessed using an *in vitro* injury assay, culturing cells in either DFM (DMEM/F12+N1) or "stress media" (no glucose (NG)-DMEM), with Biotin added over a range from 2.5 to 250 μg/ml, and cell viability determined after 24 hrs. Biotin reduced the increase in OPC cell death in the NG condition. In nanofiber myelination assays, biotin increased the percentage of ensheathing cells, the number of ensheathed segments per cell, and length of ensheathed segments. In dispersed cell culture, Biotin also significantly increased ATP production, assessed using a Seahorse bio-analyzer. For most assays, the positive effects of Biotin were observed at the higher end of the dose-response analysis. We conclude that Biotin, *in vitro*, protects OL lineage cells from metabolic injury, enhances myelin-like ensheathment, and is associated with increased ATP production.

## Introduction

Multiple sclerosis (MS) most frequently follows an initial relapsing course that then evolves into a more progressive disorder. Mechanisms implicated in disease progression include continued tissue injury and failure of repair. The initial lesions of MS feature destruction of myelin with variable extent of loss of its cell of origin, the oligodendrocyte (OL). In chronic lesions, there is universal loss of OLs [1, 2]. The degree of recovery from relapses in MS following the acute event is attributed, at least in part, to remyelination dependent on recruitment of OL progenitor cells (OPCs) with differentiation into myelin-sheath forming mature OLs [3]. Recent evidence also implicates participation of existent OLs [4–6]. OPCs are increased in

**Funding:** This study was funded by a research grant to McGill University (Jack Antel, PI) from Medday Pharmaceuticals The funders had no role in study design, data collection and analysis, decision to publish, or preparation of the manuscript.

**Competing interests:** Funding from Medday Pharmaceutical does not alter our adherence to PLOS ONE policies on sharing data and materials."

periplaque white matter (PPWM) in early MS lesions, but decreased in chronic MS lesions [7, 8]. We postulated there was a differentiation block of OPCs in chronic multiple sclerosis lesions, which may contribute to failure of remyelination [8].

Causes of OL injury and loss in MS include micro-environmental conditions of metabolic stress such as ischemia/hypoxia and mitochondrial injury [9–12]. OL loss is best recognized in the pattern 3 lesion subtype [1], that features an "oligodendrogliopathy" defined by retraction of OL terminal processes ("dying back") [2]. We have observed retraction of terminal processes of OLs even in absence of cell death in the majority of early MS lesions and at the rims of chronic active lesions [13]. OL/myelin loss impacts the survival and function of neurons/axons [14]. There are as yet no agents approved for use in MS based on neuroprotective or remyelination effects.

Biotin is a co-factor required for the activity of a family of carboxylases involved in pathways that could potentially serve protective roles for cells exposed to hypoxic/ischemic conditions as implicated in MS, as well as enhance myelination (reviewed in [15]). These carboxylases include three that are central to aerobic energy production and generate intermediates for the tricarboxylic acid (TCA) cycle. Such effects are predicted to enhance ATP production. Biotin is also a co-factor for acetyl-CoA carboxylases involved in fatty acid synthesis, a process underlying myelin production. High dose Biotin is demonstrated to reverse rare genetic CNS disorders involving energy metabolism defects [16]. In an initial double-blind placebo controlled study, high dose Pharmaceutical grade Biotin, MD1003 (100 mg three times daily), achieved sustained reversal of MS-related disability in a subset of patients with progressive MS, not seen in the placebo group [17]. However the second pivotal Phase III trial (SPI2) did not meet its primary and secondary end points (Medday announcement (03/10/2020).

In previous studies, we have used in vitro systems to model the response of human and rodent OL lineage cells to ischemic/hypoxic conditions and to assess myelination capacity [13, 18–21]. For the former, we used dissociated cell cultures of such cells subjected to conditions of glucose and nutrient deprivation [2, 20–22]. We observe initial process retraction followed by cell death [22]. This metabolic insult reduces the rate of oxygen consumption and ATP production by these cells. OL lineage cells derived from postnatal rats are still at an OPC stage and are more susceptible to cell death under these conditions than are adult human brain derived mature OLs [20, 22]. The rodent cells display activated caspase 3 and a high percentage are TUNEL+ [13, 20, 21]; and this cell death can be inhibited by a pan caspase inhibitor [13]. To assess the capacity to myelinate, we used ensheathment of synthetic nanofibers, quantified using a high-throughput program with single-cell resolution [23].

In the current study, we directly examine the *in vitro* effects of high dose biotin on functional properties of post-natal rat derived OPCs as related to protection from conditions of glucose deprivation and capacity to ensheath nanofibers. We then directly assess the biotin effect on baseline oxidative metabolism and ATP production by the cells using a Seahorse XF Analyzer. In addition, we examined the expression of biotin-dependent carboxylases [15] in adult human oligodendrocytes under physiological or metabolic stress conditions (low glucose), mimicking MS lesion microenvironment.

## Materials and methods

### Cell isolation and culture

**Rodent oligodendrocyte cultures.** All procedures involving animals were performed in accordance with the Canadian Council on Animal Care's guidelines for the use of animals in research and approved by the McGill University Animal Care Committee. OPCs were prepared from the brains of newborn Sprague-Dawley rats (purchased from Charles River, Saint

Constant, Canada) as previously described [24]. Microglia were removed by an initial shake-off, total cells in flasks were digested by trypsin, and the cells were selected using magnetic beads conjugated with monoclonal antibody A2B5, which recognizes a cell surface ganglioside [25] to select a progenitor cell pool that comprises ~30% of the total cells. OPCs were initially plated at a density of $2.5 \times 10^5$ cells per mL on poly-lysine-coated chamber slides, and cultured in defined medium (DFM) consisting of Dulbecco's Modified Eagle Medium/Nutrient Mixture F-12 (DMEM-F12) supplemented with N1, 0.01% bovine serum albumin (BSA), 1% penicillin-streptomycin, and B27 supplement (Invitrogen, Burlington, ON), platelet derived growth factor (PDGF)-AA (10ng/ml), and basic fibroblast factor (bFGF, 10ng/ml) (Sigma, Oakville, ON). Culture media was changed every 48 hrs under the stated conditions. >80–90% % of cells were O4+ after the initial 4 days in culture following cell isolation.

**Proliferation and protection assays.** For injury assays, cells were cultured in either DFM (DMEM/F12+N1) or "stress media" (no glucose (NG)-DMEM). OPCs were treated for 24 hrs with the indicated concentrations of high dose Pharmaceutical grade Biotin (MD1003), dissolved in PBS. Cell viability was assessed after 24 hrs by live-staining with propidium iodide (PI) (1:200, 15 min, 37˚ C, Invitrogen, Burlington, ON). To identify OPCs, cells were incubated with monoclonal O4 antibody (IgM, 1:200, R&D Systems, Oakville, ON) [26] for 30 min at 4˚C then fixed in 4% paraformaldehyde for 10 min at 4˚C, washed twice with PBS, followed by blocking with HHG (1 mM HEPES, 2% horse serum, 10% goat serum, Hanks' balanced salt solution) for 10 min. Secondary antibodies were either goat anti-mouse IgM Alexa Fluor 488 (1:500, Thermo Fisher Scientific, Eugene, OR) or goat anti-mouse IgM-Cy3 (1:250, Abcam, Toronto, ON) added for 30 min at 4˚C. Proliferating cells were identified by immunostaining with Ki67-FITC antibody (1:200 dilution, 4˚ C, overnight, Cell Signaling Technology). Monoclonal antibodies against galactocerebroside (GC) ($IgG_3$ 1:50, 30 min, 4˚ C, derived from hybridoma, [27]) and myelin basic protein (MBP) ($IgG_{2b}$, 1:500 dilution, 4˚ C, overnight, Sternberger, Lutherville, MD) were used to stain the cells to determine OPC differentiation after 24 hrs and 3 days of biotin treatment, respectively. The corresponding secondary antibodies were Goat anti-mouse $IgG_3$-FITC (1:100, 1 hour, room temperature, Molecular Probes, Eugene, OR) and Goat anti-mouse $IgG_{2b}$-FITC or TxR (1:100, 1 hour, room temperature, Biosource, Camarillo, CA). Cell nuclei were stained with Hoechst 33258 (10 µg/ml, Invitrogen) for 10 min at room temperature. Cells were then imaged via an epifluorescent microscope (Zeiss).

PI+ and Ki67+ cells were analyzed using a MATLAB program. The script first identified all circular Hoechst+ nuclei using the extended local minima of the distance transformed binary image for watershed segmentation. A similar procedure is used to identify PI+ and Ki67 + blebs in their respective channels. All circular blebs were then subjected to a circularity-metric to ensure exclusion of non-circular artifacts and only circular nuclear-like structures retained. Finally, Hoechst+ nuclei were co-localized with PI+ and Ki67+ blebs on a per pixel basis, and the number of positive cells counted. PDGFAA/bFGF was used as positive control.

**Nanofiber ensheathment assay.** OPCs were plated in multi-well aligned Nanofiber plates (The Electrospinning Company Ltd., Didcot, Oxfordshire, OX11 0RL) and treated with biotin for 3 days (earliest time to observe nanofiber ensheathment by OPCs—see Xu et al [23]), followed by immunostaining with O4 primary antibody and corresponding secondary antibodies conjugated with either Alexa Fluor 488 (1:500, Thermo Fisher Scientific, Eugene, OR) or Texas Red (1:100, Biosource, Camarillo, CA). Images were acquired with a Zeiss fluorescence microscope, % ensheathment by O4+ cells was quantified using MATLAB software as described by Xu et al [23]. BDNF and PDGFAA/bFGF were used as positive and negative controls respectively.

**Metabolic analysis.** OPCs were analyzed after first being grown in DFM for 4 days and then treated with biotin for 1 day. Metabolic measurements were carried out as described

previously [22]. The cells were washed with XF assay medium (pH adjusted to 7.4) and equilibrated for 1 hr in the Seahorse incubator. The XF96 plate was inserted into the Seahorse analyzer (Seahorse Bioscience, Billerica, MA) where 4 basal assay cycles were performed consisting of a 3 min mix followed by 3 min measure. After completion, oligomycin (OLIGO, 0.5 μM), a mitotoxin that inhibits ATP synthase (complex V), decreasing electron flow through the electron transport chain (ETC) with resultant reduction in mitochondrial respiration or Oxygen consumption rate (OCR), was added by automatic pneumatic injection for 3 assay cycles. Carbonyl cyanide-4-(trifluoromethoxy) phenylhydrazone (FCCP, 0.5 μM), an uncoupling agent that collapses the proton gradient and disrupts the mitochondrial membrane potential, was then added for an additional three assay cycles followed by rotenone (R, 0.1μM), a complex I inhibitor, plus antimycin A (AA, 0.2μM), a complex III inhibitor, for another three assay cycles. This combination (R/AA) shuts down mitochondrial respiration and enables the calculation of nonmitochondrial respiration driven by processes outside the mitochondria (Agilent, manufacturer's instructions). The coupling efficiency to calculate mitochondrial ATP production was derived from differences in oxygen consumption rate (OCR) upon addition of oligomycin compared to basal rate, converting OCR to ATP production using a phosphate/oxygen ratio of 5.5 [28, 29]. Extracellular acidification rates (ECAR) were calculated by the addition of 2-deoxy-glucose (2DG; 1M). Proton production rate (PPR) was utilized to estimate ATP production from glycolysis in a 1:1 ratio [30]. All reagents were purchased from Sigma (St. Louis, MO).

**Gene expression.** Human adult brain derived OL expression of biotin-dependent carboxylases–these data were derived from a previously reported microarray analysis performed on human adult brain derived OLs maintained in dissociated culture under control or 48 hrs of glucose/nutrient deprivation conditions [18].

**Statistical analysis.** All statistical analyses were performed with GraphPad Prism. 1 way-ANOVA was performed followed by Dunnett's multiple comparison Test for Fig 1A and 1B, 2A, and 3E. For Fig 2A, the % of ensheathed O4+ cells was compared between the DFM control and each concentration of biotin by paired T-test. For Fig 2B, 1 way-ANOVA (Kruskal-Wallis test) was performed followed by Dunnett's multiple comparison Test. For Fig 2C, unpaired T-test was performed to compare the control and each concentration of biotin. For Fig 3C, we compared the means of OCR for each seahorse treatment (basal, oligomycin (OLIGO), FCCP, R/AA and 2DG) between control vs 250 μg/ml, control vs. 25 μg/ml, using paired t-test. For Fig 3B, 2 way-ANOVA was performed. * $P < 0.05$, ** $P < 0.01$, *** $P < 0.001$.

## Results

Biotin mediated protection from "stress" conditions—Under glucose-free (NG) conditions, the mean % cell death (PI+ OPCs) at 24 hrs was 29 ± 7% (11–68%) vs. 8± 2% (2,5–12%) for control conditions (n = 8, p = 0.02, S1 Fig). As shown in Fig 1A, supplementation with biotin was done over a concentration range of 0.25 ng/ml to 250 μg/ml; significant reductions in the % PI+ cells were noted over the 2.5μg/ml to 250 μg/ml range (mean % reduction 35 ± 5%, n = 8, p < 0.001 at 250 μg/ml). PI staining is illustrated in Fig 1C–1F. Biotin had no significant effects on protection of OPCs at concentrations of 0.25 ng/ml to 1.25 μg/ml. Biotin supplementation did not induce a significant change in cell proliferation as measured by % Ki67 + cells (Fig 1B), at any of concentrations used. The % of Ki67+ cells is 16.1±2.6% under basal DFM conditions. Ki67 staining is illustrated in Fig 1G–1J. Cultures supplemented with PDGFAA and bFGF were used as the positive control in the proliferation assay. There was a trend for Biotin to increase GC+ cells after 1 day treatment (1.4 ± 0.2 fold of control by 250μg/ml biotin, n = 8, p = 0.07 S2 Fig). The % of MBP+ cells was not increased after 3 days treatment.

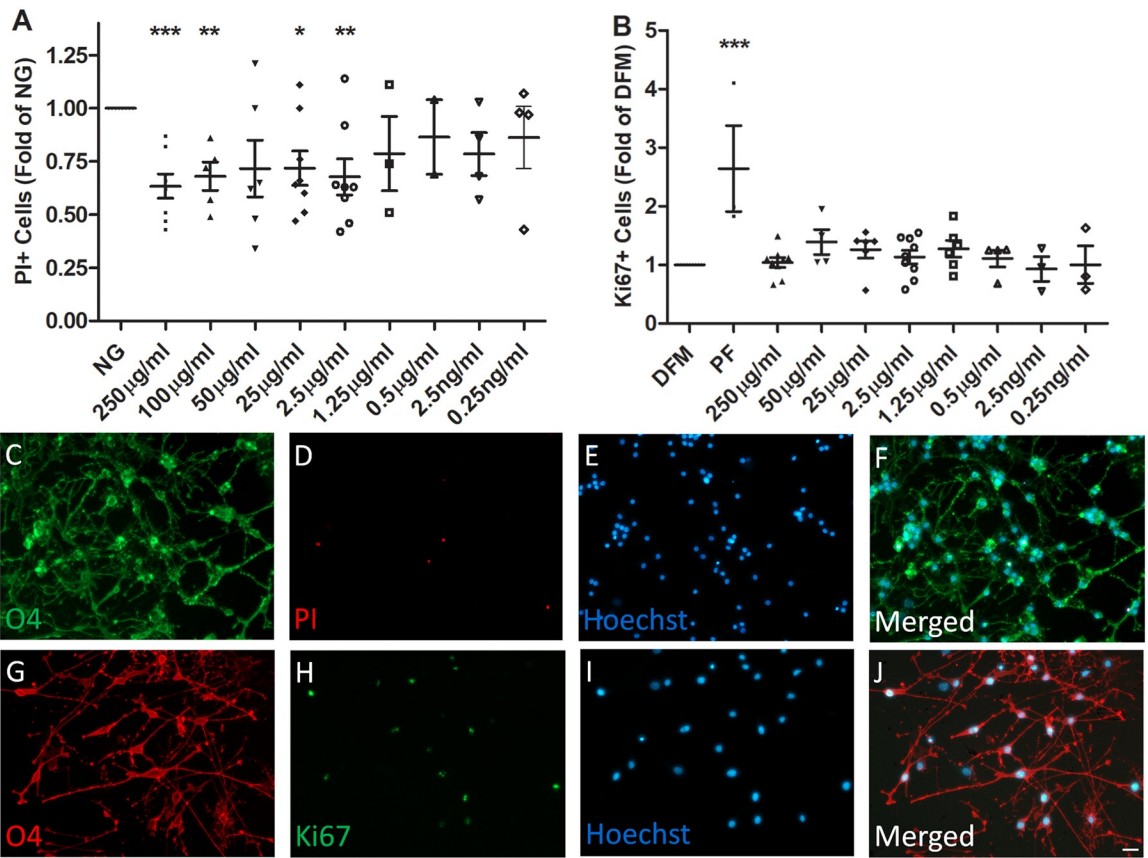

**Fig 1. Effects of biotin on OPC survival and proliferation.** Panel A, OPCs were treated with indicated concentrations of biotin when cultured in glucose-free DMEM (NG) for 24h. For this analysis, OPCs were immunostained with monoclonal antibody O4 and PI (panels C-F). Cells were imaged and PI+ cells were analyzed using a MATLAB program as described in the methods. Panel B, OPCs were treated with indicated concentrations of biotin when cultured in DFM for 24h. For this analysis, OPCs were immunostained with O4 and Ki67 (panels G-J). Cells were imaged and Ki67+ cells were analyzed using a MATLAB program described in the methods. PF: PDGFAA + bFGF. N = 8. Panel A—% PI + OPCs in presence of biotin compared to NG conditions alone; Panel B—% Ki67+ OPCs in presence of biotin or PF alone compared to DFM alone.(paired t-test): *, p < 0.05; **, p < 0.01; ***, p < 0.001. Scale bar = 10 μm.

Biotin mediated enhancement of nanofiber ensheathment under DFM condition–Overall biotin supplementation increased the % of nanofiber ensheathing O4+ cells compared to defined media (DFM) control conditions over the 2.5–250 μg/ml dose range (27.58±6.25 for 250 μg/ml biotin vs. 16.31±4.54 DFM, n = 7, p <0.05) (Fig 2A). At the individual cell level, biotin supplementation also increased the number of ensheathed segments per cell (panel 2B), and the proportion of cells with higher log length of segments (panel C). As expected, BDNF (10 ng/ml) enhanced ensheathment whereas PDGFAA/bFGF was inhibitory. Examples of nanofiber ensheathment under control and biotin supplemented conditions are provided in Fig 2D and 2E, respectively. Almost all cells are O4+ but only a small minority (<5%) are MBP+.

Biotin increased ATP production by OPCs under DFM control conditions. As shown in Fig 3A and 3B, OCR under all conditions (basal and after addition of mitotoxins) was higher for cells supplemented with biotin at 250 and 25 μg/ml compared to control conditions. P values are shown in Panel C. ECAR were at the margin of detection and no biotin-related effect could be observed (S3 Fig). In order to determine the contribution of non-mitochondrial,

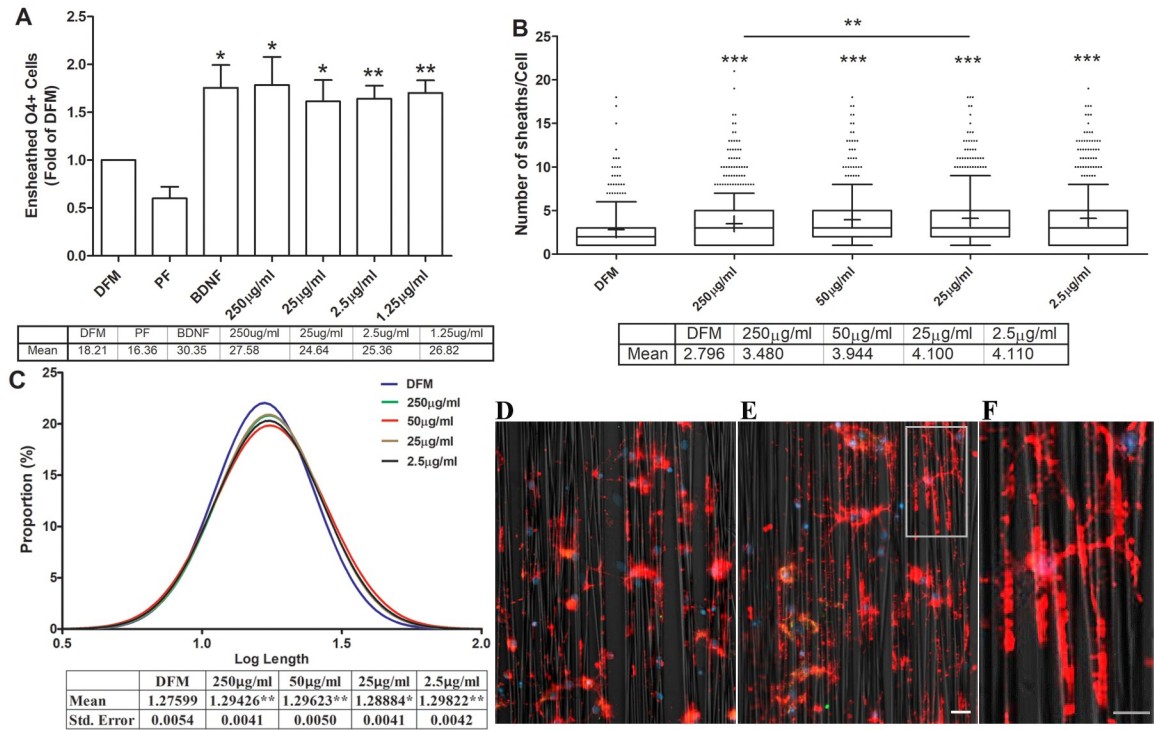

**Fig 2. Effects of biotin on OL ensheathment.** OPCs growing on nanofibers were treated with indicated concentrations of biotin in DFM for 3 days. The cell cultures were immunostained with O4 (red) and MBP (green) antibodies. Cells were imaged. % of ensheathed O4+ cells (panel A), number of sheaths per cells (panel B) and the proportion of length of sheath segment per cell (panel C) were analyzed using a MATLAB program and illustrated (mean and/or SEM for each condition are listed in the table underneath each graph). Panels D and E are examples of cells under DFM alone and biotin (250 μg/ml) treated conditions, respectively. Panel F is a higher magnification of a cell from panel E. N = 7. Comparison to control DFM (pair t-test): *, p < 0.05; ** p < 0.01, *** p < 0.001. Scale bars:10 μm in panel E, 20 μm in panel F.

ATP-linked, and proton leak components to overall OCR, we expressed each as a % of total OCR for each condition, as shown in panel D. Biotin did not change non-mitochondria oxygen consumption that accounts for ~50% of the OCR. Biotin did significantly increase the ATP-linked contribution to OCR at the higher concentration (250 μg/ml) as shown in Panel E (the calculation is described in method). The calculated ATP-linked OCR production was 6.0 ±0.22 pmol/min/μg protein with 250 μg/ml biotin vs. 2.7±0.61 in DFM conditions alone (n = 4, p < 0.01).

Expression of Biotin regulated carboxylase genes in human OLs subjected to glucose/nutrient deprivation–Table 1 shows a suggestive decrease of biotin-regulated carboxylase genes except for MCCC1 in adult human OLs cultured for 2 days in the deprivation conditions. At this time there was minimal cell death and only initial indication of process retraction as shown in our previous study [13].

## Discussion

The current studies demonstrate the measurable capacity of high dose Biotin to protect OL lineage cell from metabolic stress, enhancing nanofiber ensheathment, and increasing ATP production. Our *in vitro* studies are based on the use of post-natal rat derived OL linage cells, obtained using a shake off method and then further enrich for the population using A2B5 antibody coated immunomagnetic beads. As expected, the majority of cells obtained, from the

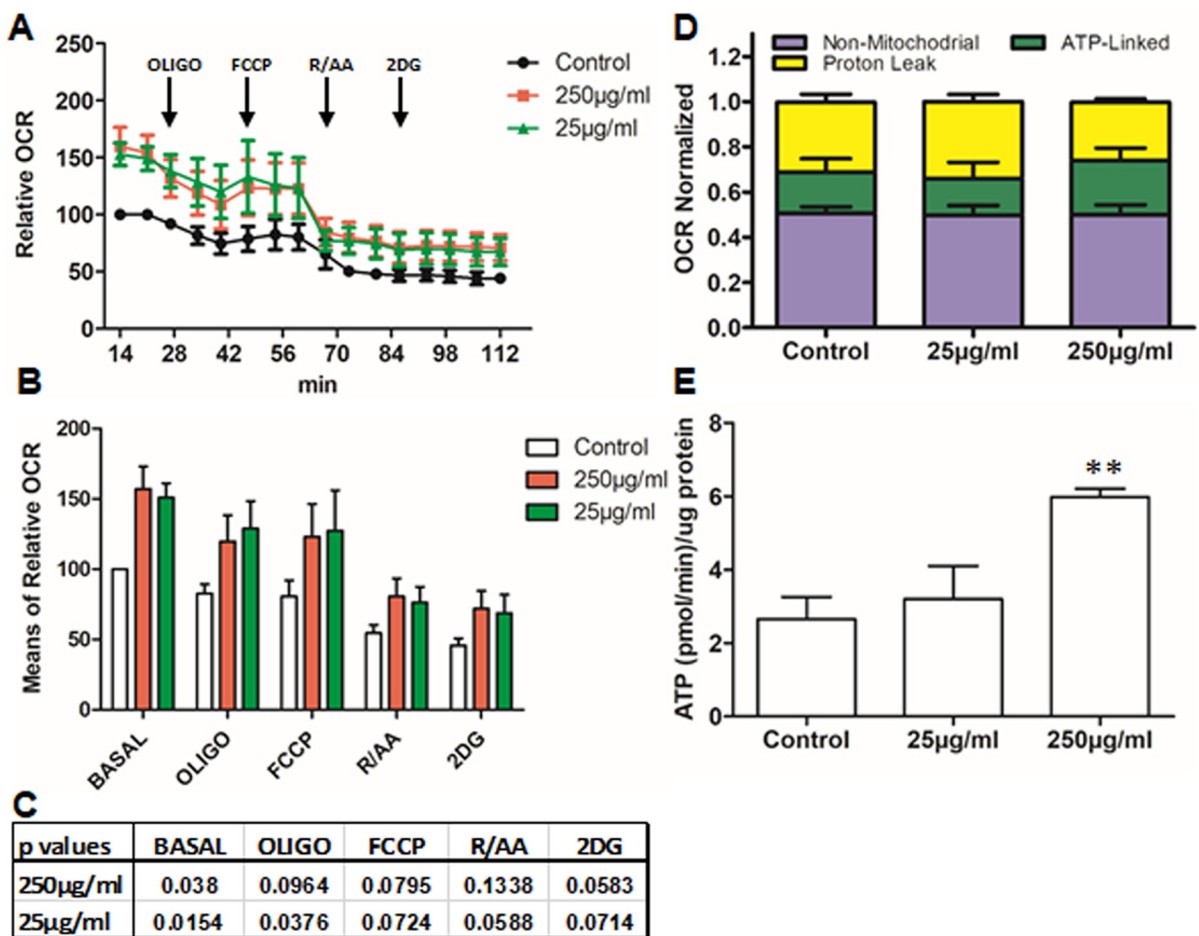

**Fig 3. Effects of biotin on OPC metabolism.** OPCs growing on seahorse plates were treated with indicated concentrations of biotin in DFM for 24 hr, followed by Seahorse analyzer measurement. Panel A presents OCR data normalized to basal values under DFM control conditions that ranged from 2.2 to 4.8 pmol O2/min/μg protein. The time point for applying OLIGO, FCCP, R/AA and 2DG is indicated. Panel B shows the mean relative basal OCR and after addition of mitotoxins. Panel C shows the p values for Biotin compared with DFM control for Panel B. Panel D shows OCR normalized; panel E shows calculated ATP production, **, p < 0.01 for 250 μg/ml Biotin compared to DFM or 25 μg/ml Biotin (1 way-ANOVA followed by Tukey multiple comparison test, n = 4).

**Table 1. Biotin regulated carboxylase gene expression in human OLs.**

| Carboxylases | AveExpr (LG) | AveExpr (N1) | geneSymbols | LogFC | P.Value |
|---|---|---|---|---|---|
| Acetyl-Coa Carboxylase A | 7.5 | 7.7 | ACACA | -0.223 | 0.24 |
| Acetyl-Coa Carboxylase B | 5.9 | 6.0 | ACACB | -0.071 | 0.66 |
| Methylcrotonyl-CoA carboxylase 1 | 6.8 | 6.6 | MCCC1 | 0.097 | 0.52 |
| Methylcrotonyl-CoA carboxylase 2 | 5.2 | 5.5 | MCCC2 | -0.271 | 0.16 |
| Pyruvate carboxylase | 6.4 | 6.6 | PC | -0.168 | 0.28 |
| Propionyl-Coa carboxylase A | 6.7 | 7.0 | PCCA | -0.494 | 0.01* |
| Propionyl-Coa carboxylase B | 7.0 | 7.3 | PCCB | -0.285 | 0.23 |

outset, are in the OL lineage as assessed by expression of O4; not many are yet MBP positive. Our basal culture media, DMEM F12, contained 0.0035mg/L of biotin https://www.thermofisher.com/ca/en/home/technical-resources/media-formulation.55.html. We did not assess biotin free media as our focus was on effects of high dose biotin modeling the *in vivo* effects of this approach.

We elected to assess dose-dependent protective effects of biotin using glucose depletion as the insult. These conditions model those that can occur in the MS lesion microenvironment [31–35]. Our previous *in vitro* studies indicate that these can induce a dying back response of OLs that may be reversible or preventable by protective strategies [13]. We had also shown using human adult brain derived OLs that this metabolic insult down-regulated pathways designated as cell morphology, cell metabolism, and cell signaling, as well as pathways specifically implicated in myelination [13]. We recognize that the OPCs used as the targets in this study may be more susceptible to the insults used compared to more mature cells derived *in vitro* from these OPCs or directly from the adult CNS [20]. OPCs injury may itself be directly relevant to the evolution of the MS disease process as such damage in MS lesions could contribute to impaired remyelination as considered below [36].

Enhancing remyelination remains a central challenge for MS with most efforts beginning with screening for agents that promote ensheathment as we have done in this study [37, 38]. Most implicate this process as being dependent on OPCs that need to survive, proliferate, migrate, differentiate, and then contact and ensheath their targets, an energy-consuming process [39]. The use of artificial nanofibers provides a convenient platform to assess ensheathment although lacking complete features of myelination such as formation of specialized paranodal loops [40, 41]. Artificial fiber assays have been used to screen for agents that promote myelination [42, 43]. The changes in the number of ensheathing cells we observed with biotin are unlikely to be related to cell proliferation/differentiation as we did not observe differences in cell numbers in our dissociated cell cultures and no differences in % O4 (or MBP) + cells. Myelination and remyelination depend on fatty acid synthesis by oligodendrocytes. Biotin is required to synthesize fatty acids and membrane formation [44], therefore we did not use biotin-free medium in our experiments. Although we did not specifically measure the expression or activity of the acetyl-coA carboxylase which could be involved in myelination [15, 45, 46], the increase in number of ensheathed segments per cell and changes in sheath length would be consistent with enhanced lipid synthesis.

To evaluate whether the increase in the number of ensheathed segments was associated with changes in energy metabolism, we assessed the effects of biotin on the metabolic status of cells under DFM basal culture conditions using a Seahorse XF bio-analyzer. As confirmed in the current study, post-natal rat OPCs mainly utilize oxidative phosphorylation to produce ATP [20, 22]. The OCR levels recorded in the current study are characteristic and comparable to our previously published respiration profiles of post-natal rat brain derived oligodendrocytes [20, 22]. The mechanisms whereby Biotin supplementation increases the resting OCR could include biotin being a co-factor of carboxylases. Three out of five biotin dependent mammalian carboxylases lie in mitochondria (Pyruvate carboxylase, Propionyl-CoA carboxylase and 3-methylcrotonyl-CoA carboxylase) [15] and have a role in providing intermediates to the TCA cycle. Increased processing of intermediates through the TCA cycle generates more NADH and FADH2. Increased NADH and FADH2 levels necessitate processing through oxidative phosphorylation accounting for OCR increment. Oxygen consumption has been shown to reflect NADH turnover in brain tissue [47]. Changes in biotin availability in yeast have been shown to impact OCR levels, at concentration of 200 μg/L [48]. As shown in rat oligodendroglia, carboxylase activities are sensitive to levels of biotin availability [49]. Further, biotin can attenuate oxidative stress and mitochondrial dysfunction in murine oligodendrocytes [50].

Our findings that biotin-dependent carboxylase gene expression level is reduced in human OLs exposed to metabolic stress conditions that have been used to model what is encountered in MS lesions [22, 51–53] suggest a basis whereby biotin may target the MS disease process within the CNS.

## Conclusion

Our *in vitro* based results indicate that high dose biotin supplementation protects OL lineage cells from metabolic injury and enhances their ensheathment capacity; such effects are associated with enhanced ATP production.

## Supporting information

**S1 Fig. Glucose-free (NG) conditions trigger cell death in OPCs.** OPCs were cultured in DFM control or NG conditions for 24 hrs. OPCs were immunostained with monoclonal antibody O4 and PI. PI+ cells were analyzed using a MATLAB program described in the methods. Comparison between DFM and NG was performed by paired t-test, p = 0.02, N = 8.
(TIF)

**S2 Fig. Effects of biotin on OPC differentiation.** Panel A, OPCs were treated with indicated concentrations of biotin in DFM for 24 hrs. OPCs were immunostained with monoclonal antibodies O4 and GC. Cells were imaged and GC+ cells were analyzed using a MATLAB program described in the methods. Panel B, OPCs were treated with indicated concentrations of biotin in DFM for 3 days. OPCs were immunostained with monoclonal antibodies O4 and MBP. 1-way ANOVA was performed followed by Dunnett's multiple comparison test.
(TIF)

**S3 Fig. Effects of biotin on OPC metabolism.** ECAR data normalized to basal values under control conditions that ranged from 0.03 to 0.3 pmol O2/min/µg protein. The time point for applying OLIGO and 2DG is indicated.
(TIF)

## Author Contributions

**Conceptualization:** Qiao-Ling Cui, Vijayaraghava T. S. Rao, Timothy E. Kennedy, Jack Antel.

**Data curation:** Qiao-Ling Cui, Yu Kang T. Xu, Milton G. F. Fernandes.

**Formal analysis:** Qiao-Ling Cui, Yun Hsuan Lin, Yu Kang T. Xu.

**Funding acquisition:** Jack Antel.

**Investigation:** Qiao-Ling Cui.

**Methodology:** Qiao-Ling Cui.

**Project administration:** Qiao-Ling Cui, Vijayaraghava T. S. Rao, Jack Antel.

**Software:** Yu Kang T. Xu.

**Supervision:** Jack Antel.

**Validation:** Qiao-Ling Cui.

**Visualization:** Qiao-Ling Cui, Yun Hsuan Lin.

**Writing – original draft:** Qiao-Ling Cui, Jack Antel.

**Writing – review & editing:** Qiao-Ling Cui, Vijayaraghava T. S. Rao, Timothy E. Kennedy, Jack Antel.

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
