## [Decision Letter · Decision Letter 0]

5 Feb 2020

PONE-D-19-34914

Effects of Biotin on survival, ensheathment, and ATP production by oligodendrocyte lineage cells in vitro

PLOS ONE

Dear Dr. Cui,

Thank you for submitting your manuscript to PLOS ONE. After careful consideration, we feel that it has merit but does not fully meet PLOS ONE’s publication criteria as it currently stands. Therefore, we invite you to submit a revised version of the manuscript that addresses the points raised during the review process.

We would appreciate receiving your revised manuscript by Mar 21 2020 11:59PM. To enhance the reproducibility of your results, we recommend that if applicable you deposit your laboratory protocols in protocols.io, where a protocol can be assigned its own identifier (DOI) such that it can be cited independently in the future. For instructions see: http://journals.plos.org/plosone/s/submission-guidelines#loc-laboratory-protocols

We look forward to receiving your revised manuscript.

Kind regards,

Ken Arai

Academic Editor

PLOS ONE

Journal Requirements:

2. Please check that the reference to previous work that described biotin-dependent carboxylase expression in human-adult-brain-derived OLs is correct. This work seems to be cited as both reference 13 and reference 18 in the text.

3. Thank you for providing the following Funding Statement: 

"Yes, this study was funded by Medday Pharmaceuticals SA, who also provided the Biotin (MD1003), the same product that is being used in their clinical trial program. The data were generated independently of Medday although by agreement was reviewed by the company."

We note that one or more of the authors is affiliated with the funding organization, indicating the funder may have had some role in the design, data collection, analysis or preparation of your manuscript for publication; in other words, the funder played an indirect role through the participation of the co-authors.

If the funding organization did not play a role in the study design, data collection and analysis, decision to publish, or preparation of the manuscript and only provided financial support in the form of authors' salaries and/or research materials, please review your statements relating to the author contributions, and ensure you have specifically and accurately indicated the role(s) that these authors had in your study in the Author Contributions section of the online submission form. Please make any necessary amendments directly within this section of the online submission form.  Please also update your Funding Statement to include the following statement: “The funder provided support in the form of salaries for authors [insert relevant initials], but did not have any additional role in the study design, data collection and analysis, decision to publish, or preparation of the manuscript. The specific roles of these authors are articulated in the ‘author contributions’ section.”

If the funding organization did have an additional role, please state and explain that role within your Funding Statement. 

Please also provide an updated Competing Interests Statement declaring this commercial affiliation along with any other relevant declarations relating to employment, consultancy, patents, products in development, or marketed products, etc. 

4. Please provide copyright information for table 1, as the data presented therein seem to be derived from prev publ from this group (https://doi.org/10.1002/ana.24944).

Reviewers' comments:

Reviewer's Responses to Questions

**Comments to the Author**

1. Is the manuscript technically sound, and do the data support the conclusions?

Reviewer #1: Partly

Reviewer #2: Yes

2. Has the statistical analysis been performed appropriately and rigorously? 

Reviewer #1: Yes

Reviewer #2: No

3. Have the authors made all data underlying the findings in their manuscript fully available?

Reviewer #1: No

Reviewer #2: Yes

4. Is the manuscript presented in an intelligible fashion and written in standard English?

Reviewer #1: No

Reviewer #2: Yes

5. Review Comments to the Author

Reviewer #1: In the manuscript entitled “Effects of Biotin on survival, ensheathment, and ATP production by oligodendrocyte lineage cells in vitro” by Qiao-Ling Cui, Yun Hsuan Lin, Yu Kang T. Xu, Milton Fernandes, Vijayaraghava TS Rao, Timothy E Kennedy, and Jack Antel, the authors assessed the effect of biotin on oligodendrocyte lineage cell (OPC), and concluded that biotin can be decrease the ratio of cell death of OPC, and promote ensheathment of myelin, and ATP synthesis in vitro assay. In addition, the authors examined gene expression analysis using data from human adult brain oligodendrocyte, and presented the list of carboxylase genes which biotin regulated in human OL.

First of all, while I appreciate the effort of this experiment, I have no idea whether there is a relationship between the results of in vitro assay from rat OPC and that of genes from human OL in this study. I agree that the authors research points are important to resolve how oligodendrocyte or OPC regulate the mechanism(s) to recover from injuries like MS and other white matter injuries. But in this paper, the authors did not show in detailed explanation in the “Results” and “Discussion” about human, so I could not imagine the relationship between both. If the authors want to include this argument, please add more detailed explanations in “Introduction”, “Results”, and “Discussion”. Or, the authors should analyze gene expression using rat OPC after biotin treatment, and you can compare the results from rat and human, and discuss this point.

In the second point, the authors mentioned that high concentration of biotin was effective on cell viability, ensheathment, and metabolic activity. But in the Fig.2B and C, it seemed that the concentration of 2.5 µg/ ml was more effective than the highest concentration (250 µg/ ml). What is the most effective concentration of biotin? And also, the authors examined the effect of 2 concentrations (25 µg/ ml and 250 µg/ ml) of biotin in Fig.3. Why did the authors choose these 2 concentrations to analyze metabolic activities? In Fig.3, it seemed that concentration of 25 µg/ ml of biotin was more effective than that of 250 µg/ml after oligomysin treatment. In Fig.3E, however, the concentration of 250 µg/ ml was effective than that of 25 µg/ ml on ATP production. I was so confused because I have no idea what the best concentration was to discuss what the authors wanted to say. Please explain more clearly in this point, and it is helpful for readers to understand this study. Also there seem some more points that need to be addressed by the authors. Please see below.

Comments;

1. In the “Materials and Methods”, the authors used some media in this study. Are there some differences in defined medium (DFM) and optimal media? If so, please unify the notation. If not, please write differences more clearly.

2. Please describe procedure of ICC; especially, dilution ratio, incubation time, washing conditions, and so on.

3. As discussed above, please describe more detailed explanation in “Gene expression”.

4. P.9, 1st paragraph, the authors said the cell death ratio under the glucose-free conditions. Please show us the data (e.g. graph). And also, the authors mentioned that biotin did not increase the MBP-positive cells, though GC-positive cells were increased after biotin treatment (P.9). Generally speaking, we know that the expression of GC is earlier than that of MBP in myelination processes. The authors assessed ICC at 24 h after biotin treatment, so it was easy to imagine that results. I have no idea why the authors mentioned.

5. P.10, 1st paragraph, the authors used the word “mitotoxins”. The authors should write all reagents name, or you should define what mitotoxins are. It is helpful for all readers to understand. Similarly, the authors wrote the word “ECAR rates”. This expression is not correct, because ECAR is an abbreviation for “ExtraCellular Acidification Rates”. In P.8, 1st paragraph, the authors wrote “extracellular acidification rate”, so please define here.

6. Please check your manuscript carefully. I can’t understand in some sentences in your manuscript.

Minor points;

- In Fig.1, 2, and supplementary Fig.1, use correct unit in your graph. Change “ug/ml” to “¨µg/ml”.

- In Fig. 3, R/AA is not common word for all readers. Please define an abbreviation in “Materials and Methods”(P.7).

- There are some typos. So please check the words carefully, e.g. KI-67, DMF.

Reviewer #2: In the context of multiple sclerosis, the authors study the influence of a high dose of biotin on oligodendrocyte precursor cells (OPCs) under metabolic stress (low-glucose concentration in the medium). The OPC cell death is increased in this condition, but the increase can be reduced in presence of a high concentration of biotin. The authors also used nanofiber myelination assays to show that biotin increases the percentage of ensheathing cells, the number of ensheathed segments per cell, and the length of ensheathed segments. They also show that in cell culture, biotin increases oxygen consumption rate and ATP production.

The article is well written and interesting.

Here are my remarks:

1- There are several names that should be explained, so that non-specialists could fully understand the paper:

- A2B5

- what are mitotoxins and what are their effects?

- same question with oligomycin.

- In the Materials and Methods section, please give the concentration of antibodies in the Proliferation and Protection assays and the Nanofiber ensheathment assay paragraphs, so that the experiments could be reproduced.

2. Figures

In Figure 1 and 2, the conditions significantly different from the control do not apple clearly. For Example, in Figure 1A: it seems that only the concentrations of 250 and 2.5 ug/mL are significantly different from the control (the only two points with an asterisk). However, at the beginning of the Result Section, it is written: "As shown in Figure 1A, supplementation with biotin over a concentration range of 2.5 to 250 μg/ml significantly reduced the % PI+ cells (mean % reduction 35+-5 %, n = 8, p < 0.001 at 250 μg/ml).

An asterisk should be put above all the data points, or the sentence should be changed.

- Same remark for Figure 2A: is only the 250 ug/mL condition statistically significant? If this is the case, it should be mentioned in the text.

- Figure 2C: I do not see the experimental data points, it seems that it is only a fit that is showed here. Please add the experimental data points, so that I could see the error bars. How are estimated the error bars on the proportion of length of sheath segment per cell? are the differences statistically significant?

- Figure 2 D and E: the figures of cells on nanofibers are not very clear, the contrast is not the same and it is really difficult to see a difference. Would it be possible to show a magnified image so that we could clearly see a few ensheathed nanofibers?

3. Metabolism

In Rao et al, PlosOne 2017 (from the same group), it is said that "adult rat oligodendrocytes preferentially use glycolysis whereas [..] oligodendrocyte progenitor cells from which they are derived, mainly use oxidative phosphorylation to produce ATP". In this article, OPCs from the brains of newborn Sprague-Dawley rats were used, as well as human adult brain derived OLs. These two types of cells should have a different metabolism. How does metabolism (glycolysis or oxidative phosphorylation) influence the effect of biotin?

6. PLOS authors have the option to publish the peer review history of their article (what does this mean?). If published, this will include your full peer review and any attached files.

Reviewer #1: No

Reviewer #2: No

---

## [Author Response · Author response to Decision Letter 0]

14 Apr 2020

Ken Arai

Academic Editor

PLOS ONE

Dear Dr Arai

We are submitting our revised manuscript which we hope you will find addresses the issues and critiques raised by the reviewers. We also address the journal requirements that you outline .Below are our specific responses to all of the above. 

Sincerely,

Jack Antel for all the authors 

PS As we were preparing to submit the revised manuscript, Medday announced that their current phase 3 high dose Biotin trial did not meet its end points. We added as sentence to the manuscript (there is as yet no scientific presentation of the data).

PLOS ONE related requirements:

Please ensure that your manuscript meets PLOS ONE's style requirements, including those for file naming. The PLOS ONE style templates can be found at http://www.plosone.org/attachments/PLOSOne_formatting_sample_main_body.pdf

 We have implemented the above recommended format.

and http://www.plosone.org/attachments/PLOSOne_formatting_sample_title_authors_affiliations.pdf

 We have implemented the above recommended format.

2. Please check that the reference to previous work that described biotin-dependent carboxylase expression in human-adult-brain-derived OLs is correct. This work seems to be cited as both reference 13 and reference 18 in the text.

 References 13 or 18 (regarding adult human brain derived oligodendrocytes) do not pertain to biotin dependent carboxylases. Reference 15 has the information regarding carboxylases. 

3. Thank you for providing the following Funding Statement: 

"Yes, this study was funded by Medday Pharmaceuticals SA, who also provided the Biotin (MD1003), the same product that is being used in their clinical trial program. The data were generated independently of Medday although by agreement was reviewed by the company."

We note that one or more of the authors is affiliated with the funding organization, indicating the funder may have had some role in the design, data collection, analysis or preparation of your manuscript for publication; in other words, the funder played an indirect role through the participation of the co-authors.

If the funding organization did not play a role in the study design, data collection and analysis, decision to publish, or preparation of the manuscript and only provided financial support in the form of authors' salaries and/or research materials, please review your statements relating to the author contributions, and ensure you have specifically and accurately indicated the role(s) that these authors had in your study in the Author Contributions section of the online submission form. Please make any necessary amendments directly within this section of the online submission form. Please also update your Funding Statement to include the following statement: “The funder provided support in the form of salaries for authors [insert relevant initials], but did not have any additional role in the study design, data collection and analysis, decision to publish, or preparation of the manuscript. The specific roles of these authors are articulated in the ‘author contributions’ section.”

We have modified our statement in the manuscript regarding support received from Medday for the study. Revised funding statement – this study was funded by a research grant to McGill University (Jack Antel, PI) from Medday Pharmaceuticals. 

 Author VR’s main contributions to this study occurred while he was a Research Associate at McGill. He joined Medday as the manuscript was being prepared. Medday employees played a consultative role during the under-taking of this project.

If the funding organization did have an additional role, please state and explain that role within your Funding Statement. No other role.

Please also provide an updated Competing Interests Statement declaring this commercial affiliation along with any other relevant declarations relating to employment, consultancy, patents, products in development, or marketed products, etc. No competing interests to declare. 

As recommended, we have included the competing interests statement as per PLOSone stipulation "This does not alter our adherence to PLOS ONE policies on sharing data and materials.”

 No competing interests to declare. Author contribution section has been updated to address potential questions regarding this.

4. Please provide copyright information for table 1, as the data presented therein seem to be derived from prev publ from this group (https://doi.org/10.1002/ana.24944). 

The data depicted in Table 1 has not been published as part of https://doi.org/10.1002/ana.24944. 

Reviewers' comments:

Reviewer's Responses to Questions

Comments to the Author

1. Is the manuscript technically sound, and do the data support the conclusions?

Reviewer #1: Partly

Reviewer #2: Yes

2. Has the statistical analysis been performed appropriately and rigorously? 

Reviewer #1: Yes

Reviewer #2: No

3. Have the authors made all data underlying the findings in their manuscript fully available?

Reviewer #1: No

Reviewer #2: Yes

4. Is the manuscript presented in an intelligible fashion and written in standard English?

Reviewer #1: No

Reviewer #2: Yes

 5. Review Comments to the Author

Please use the space provided to ex \\plain your answers to the questions above. You may also include additional comments for the author, including concerns about dual publication, research ethics, or publication ethics. (Please upload your review as an attachment if it exceeds 20,000 characters) 

Our specific responses are included below and changes to the manuscript are highlighted.

Reviewer #1: In the manuscript entitled “Effects of Biotin on survival, ensheathment, and ATP production by oligodendrocyte lineage cells in vitro” by Qiao-Ling Cui, Yun Hsuan Lin, Yu Kang T. Xu, Milton Fernandes, Vijayaraghava TS Rao, Timothy E Kennedy, and Jack Antel, the authors assessed the effect of biotin on oligodendrocyte lineage cell (OPC), and concluded that biotin can be decrease the ratio of cell death of OPC, and promote ensheathment of myelin, and ATP synthesis in vitro assay. In addition, the authors examined gene expression analysis using data from human adult brain oligodendrocyte, and presented the list of carboxylase genes which biotin regulated in human OL.

First of all, while I appreciate the effort of this experiment, I have no idea whether there is a relationship between the results of in vitro assay from rat OPC and that of genes from human OL in this study. I agree that the authors research points are important to resolve how oligodendrocyte or OPC regulate the mechanism(s) to recover from injuries like MS and other white matter injuries. But in this paper, the authors did not show in detailed explanation in the “Results” and “Discussion” about human, so I could not imagine the relationship between both. If the authors want to include this argument, please add more detailed explanations in “Introduction”, “Results”, and “Discussion”. Or, the authors should analyze gene expression using rat OPC after biotin treatment, and you can compare the results from rat and human, and discuss this point.

We agree with the reviewer’s comments that in the original manuscript we did not attempt to explain the “relationship between the results of in vitro assay from rat OPC and that of genes from human OL in this study”. We now present our rationale for the human oligodendrocyte studies in the introduction and provide a statement in the discussion to show how our in vitro rat based functional studies link with the human oligodendrocyte molecular studies. 

The following sentences have been added in the introduction and discussion sections, 

Introduction: “We examined the expression of biotin-dependent carboxylases in adult human oligodendrocytes under physiological or metabolic stress conditions (low glucose), mimicking MS pathology.”

Discussion – new last sentence in discussion: As shown in rat oligodendroglia, carboxylase activities are sensitive to levels of biotin availability [49]. Further, biotin can attenuate oxidative stress and mitochondrial dysfunction in murine oligodendrocytes [50]. Our findings that biotin-dependent carboxylase gene expression levels are reduced in human OLs exposed to metabolic stress conditions that have been used to model what is encountered in MS lesions [22, 51-53]suggest a basis whereby biotin may target the MS disease process within the CNS.

In the second point, the authors mentioned that high concentration of biotin was effective on cell viability, ensheathment, and metabolic activity. But in the Fig.2B and C, it seemed that the concentration of 2.5 µg/ ml was more effective than the highest concentration (250 µg/ ml). What is the most effective concentration of biotin? 

Fig 2 in the results section has been modified. In figure 1 we showed that the protective effect of biotin is observed over the 250-2.5 ug/ml range. We now show more clearly in the revised fig 2, panel A that the ensheathment effect is also seen over this dose range )p values now included). We are less sure about distinguishing between effects within this range. Panel B already showed the significance of the Biotin effect. Panel C – we now include SEMs and p values in the legend. We have added Panel F(a higher magnification of the insert in panel E) so as to better appreciate ensheathment. 

And also, the authors examined the effect of 2 concentrations (25 µg/ ml and 250 µg/ ml) of biotin in Fig.3. Why did the authors choose these 2 concentrations to analyze metabolic activities?

We chose these because they were at the extremes of the dose range where we observed effects in figures 1 and 2. 

 In Fig.3, it seemed that concentration of 25 µg/ ml of biotin was more effective than that of 250 µg/ml after oligomysin treatment. In Fig.3E, however, the concentration of 250 µg/ ml was effective than that of 25 µg/ ml on ATP production. I was so confused because I have no idea what the best concentration was to discuss what the authors wanted to say. Please explain more clearly in this point, and it is helpful for readers to understand this study. Also there seem some more points that need to be addressed by the authors. Please see below.

In panels A and B there are marginal difference between the 2 dosages when compared to each other under all conditions. However when mitochondrial ATP-linked OCR and correlated ATP production is calculated (ie differences between oxygen consumption rate (OCR) upon addition of oligomycin and the basal OCR), there is a significant difference between the 2 dosages (higher with the higher dose) as now indicated in panel E (p values for the comparison are now given)..

Comments;

1. In the “Materials and Methods”, the authors used some media in this study. Are there some differences in defined medium (DFM) and optimal media? If so, please unify the notation. If not, please write differences more clearly. We changed the optimal media to DFM.

2. Please describe procedure of ICC; especially, dilution ratio, incubation time, washing conditions, and so on. We added the detailed information of ICC to the methods section.

3. As discussed above, please describe more detailed explanation in “Gene expression”. 

Our methods are given in our previously published paper as cited (ref 13)– we can provide these again if requested.

4. P.9, 1st paragraph, the authors said the cell death ratio under the glucose-free conditions. Please show us the data (e.g. graph). 

We show the data in the figure below – we have added it to the manuscript as supplementary figure 1. The original manuscript shows only the mean values in the results section. 

And also, the authors mentioned that biotin did not increase the MBP-positive cells, though GC-positive cells were increased after biotin treatment (P.9). Generally speaking, we know that the expression of GC is earlier than that of MBP in myelination processes. The authors assessed ICC at 24 h after biotin treatment, so it was easy to imagine that results. I have no idea why the authors mentioned MBP.

We measured MBP expression after 3 days treatment with Biotin but we did not see expression. (clarified this in the method and results sections).

5. P.10, 1st paragraph, the authors used the word “mitotoxins”. The authors should write all reagents name, or you should define what mitotoxins are. It is helpful for all readers to understand. Similarly, the authors wrote the word “ECAR rates”. This expression is not correct, because ECAR is an abbreviation for “ExtraCellular Acidification Rates”. In P.8, 1st paragraph, the authors wrote “extracellular acidification rate”, so please define here. Corrected as suggested 

6. Please check your manuscript carefully. I can’t understand in some sentences in your manuscript. 

We hope our revisions have clarified this.

Minor points;

- In Fig.1, 2, and supplementary Fig.1, use correct unit in your graph. Change “ug/ml” to “¨µg/ml”. Done.

- In Fig. 3, R/AA is not common word for all readers. Please define an abbreviation in “Materials and Methods”(P.7). Done.

- There are some typos. So please check the words carefully, e.g. KI-67, DMF. Done.

Reviewer #2: In the context of multiple sclerosis, the authors study the influence of a high dose of biotin on oligodendrocyte precursor cells (OPCs) under metabolic stress (low-glucose concentration in the medium). The OPC cell death is increased in this condition, but the increase can be reduced in presence of a high concentration of biotin. The authors also used nanofiber myelination assays to show that biotin increases the percentage of ensheathing cells, the number of ensheathed segments per cell, and the length of ensheathed segments. They also show that in cell culture, biotin increases oxygen consumption rate and ATP production.

The article is well written and interesting.

Here are my remarks:

1- There are several names that should be explained, so that non-specialists could fully understand the paper:- A2B5. 

We now specify in the methods section that A2B5 cells are recognized based on reactivity with this antibody.

- what are mitotoxins and what are their effects?

- same question with oligomycin.

Below is a detailed description from the literature. We summarize this in our revised manuscript (methods section)

Mitotoxins used in our Seahorse assay are: Oligomycin, Carbonyl cyanide-4 (trifluoromethoxy) phenylhydrazone (FCCP) and Rotenone with Antimycin A

Effects of mitotoxins: 

Oligomycin inhibits ATP synthase (complex V), and is injected first in the assay following basal measurements. It impacts or decreases electron flow through the electron transport chain (ETC), resulting a reduction in mitochondrial respiration or Oxygen consumption rate (OCR). This decrease in OCR is linked to cellular ATP production.

FCCP is an uncoupling agent that collapses the proton gradient and disrupts the mitochondrial membrane potential. It is the 2nd injection following Oligomycin. As a result, electron flow through the ETC is uninhibited, and oxygen consumption by complex IV reaches the maximum. 

The third injection is a mixture of rotenone, a complex I inhibitor, and antimycin A, a complex III inhibitor. This combination shuts down mitochondrial respiration and enables the calculation of nonmitochondrial respiration driven by processes outside the mitochondria.

Referenced from https://www.agilent.com/cs/library/usermanuals/public/XF_Cell_Mito_Stress_Test_Kit_User_Guide.pdf

- In the Materials and Methods section, please give the concentration of antibodies in the Proliferation and Protection assays and the Nanofiber ensheathment assay paragraphs, so that the experiments could be reproduced. Done as requested. 

2. Figures

In Figure 1 and 2, the conditions significantly different from the control do not apple clearly. For Example, in Figure 1A: it seems that only the concentrations of 250 and 2.5 ug/mL are significantly different from the control (the only two points with an asterisk). However, at the beginning of the Result Section, it is written: "As shown in Figure 1A, supplementation with biotin over a concentration range of 2.5 to 250 μg/ml significantly reduced the % PI+ cells (mean % reduction 35+-5 %, n = 8, p < 0.001 at 250 μg/ml).

An asterisk should be put above all the data points, or the sentence should be changed.

We have done as requested (please see detailed response to reviewer 1.

- Same remark for Figure 2A: is only the 250 ug/mL condition statistically significant? If this is the case, it should be mentioned in the text.

. We have done as requested (please see detailed response to reviewer 1).

- Figure 2C: I do not see the experimental data points, it seems that it is only a fit that is showed here. Please add the experimental data points, so that I could see the error bars. How are estimated the error bars on the proportion of length of sheath segment per cell? are the differences statistically significant?

We now provide a legend that contains mean values + SEMs + p values. Fitting error bars to the actual curves on the graphs would be difficult.

- Figure 2 D and E: the figures of cells on nanofibers are not very clear, the contrast is not the same and it is really difficult to see a difference. Would it be possible to show a magnified image so that we could clearly see a few ensheathed nanofibers?

 We added a magnified image of an individual cell in the revised figure.

3. Metabolism

In Rao et al, PlosOne 2017 (from the same group), it is said that "adult rat oligodendrocytes preferentially use glycolysis whereas [..] oligodendrocyte progenitor cells from which they are derived, mainly use oxidative phosphorylation to produce ATP". In this article, OPCs from the brains of newborn Sprague-Dawley rats were used, as well as human adult brain derived OLs. These two types of cells should have a different metabolism. How does metabolism (glycolysis or oxidative phosphorylation) influence the effect of biotin? 

 It is correct that postnatal rat OPCs use oxidative phosphorylation to a greater extent as compared to adult rat brain derived oligodendrocytes. It is possible that such cells benefit from increased biotin availability by potentially improving the mitochondrial respiration efficiency. In the manuscript we state that “Three out of five biotin dependent mammalian carboxylases lie in mitochondria (Pyruvate carboxylase, Propionyl-CoA carboxylase and 3-methylcrotonyl-CoA carboxylase).”

In the current study we only used the adult human OLs for the studies of gene expression under stress conditions as discussed in our response to reviewer 1.

6. PLOS authors have the option to publish the peer review history of their article. If published, this will include your full peer review and any attached files.

Would not wish to have reviews published 

Do you want your identity to be public for this peer review? For information about this choice, including consent withdrawal, please see our Privacy Policy.

Reviewer #1: No

Reviewer #2: No

---

## [Decision Letter · Decision Letter 1]

14 May 2020

Effects of Biotin on survival, ensheathment, and ATP production by oligodendrocyte lineage cells in vitro

PONE-D-19-34914R1

Dear Dr. Cui,

We are pleased to inform you that your manuscript has been judged scientifically suitable for publication and will be formally accepted for publication once it complies with all outstanding technical requirements.

With kind regards,

Ken Arai

Academic Editor

PLOS ONE

Additional Editor Comments (optional):

Reviewers' comments:

Reviewer's Responses to Questions

**Comments to the Author**

1. If the authors have adequately addressed your comments raised in a previous round of review and you feel that this manuscript is now acceptable for publication, you may indicate that here to bypass the “Comments to the Author” section, enter your conflict of interest statement in the “Confidential to Editor” section, and submit your "Accept" recommendation.

Reviewer #1: (No Response)

Reviewer #2: All comments have been addressed

2. Is the manuscript technically sound, and do the data support the conclusions?

Reviewer #1: (No Response)

Reviewer #2: Yes

3. Has the statistical analysis been performed appropriately and rigorously? 

Reviewer #1: (No Response)

Reviewer #2: Yes

4. Have the authors made all data underlying the findings in their manuscript fully available?

Reviewer #1: (No Response)

Reviewer #2: Yes

5. Is the manuscript presented in an intelligible fashion and written in standard English?

Reviewer #1: (No Response)

Reviewer #2: Yes

6. Review Comments to the Author

Reviewer #1: (No Response)

Reviewer #2: The authors have answered to my questions and all my comments have been addressed. The manuscript has been greatly improved.

7. PLOS authors have the option to publish the peer review history of their article (what does this mean?). If published, this will include your full peer review and any attached files.

Reviewer #1: No

Reviewer #2: No

---

## [Editor Report · Acceptance letter]

20 May 2020

PONE-D-19-34914R1 

Effects of Biotin on survival, ensheathment, and ATP production by oligodendrocyte lineage cells in vitro 

Dear Dr. Cui:

I am pleased to inform you that your manuscript has been deemed suitable for publication in PLOS ONE. Congratulations! Your manuscript is now with our production department. 

With kind regards,

on behalf of

Dr. Ken Arai 

Academic Editor

PLOS ONE